# Computational analysis on two putative mitochondrial protein-coding genes from the *Emydura subglobosa* genome: A functional annotation approach

**Megan Yu** [ORCID] *

Department of Molecular, Cell & Developmental Biology, University of California–Los Angeles, Los Angeles, California, United States of America

* yumr247@g.ucla.edu

## Abstract

Rapid advancements in automated genomic technologies have uncovered many unique findings about the turtle genome and its associated features including olfactory gene expansions and duplications of toll-like receptors. However, despite the advent of large-scale sequencing, assembly, and annotation, about 40–50% of genes in eukaryotic genomes are left without functional annotation, severely limiting our knowledge of the biological information of genes. Additionally, these automated processes are prone to errors since draft genomes consist of several disconnected scaffolds whose order is unknown; erroneous draft assemblies may also be contaminated with foreign sequences and propagate to cause errors in annotation. Many of these automated annotations are thus incomplete and inaccurate, highlighting the need for functional annotation to link gene sequences to biological identity. In this study, we have functionally annotated two genes of the red-bellied short-neck turtle (*Emydura subglobosa*), a member of the relatively understudied pleurodire lineage of turtles. We improved upon initial *ab initio* gene predictions through homology-based evidence and generated refined consensus gene models. Through functional, localization, and structural analyses of the predicted proteins, we discovered conserved putative genes encoding mitochondrial proteins that play a role in C21-steroid hormone biosynthetic processes and fatty acid catabolism—both of which are distantly related by the tricarboxylic acid (TCA) cycle and share similar metabolic pathways. Overall, these findings further our knowledge about the genetic features underlying turtle physiology, morphology, and longevity, which have important implications for the treatment of human diseases and evolutionary studies.

## Introduction

The crown group of living turtles (Testudines) comprise a relatively ancient lineage dating back 220 million years, with the earliest stem turtles extending back at least another 40 million years [1, 2]. Compared to other eukaryotic species' groups, turtles have minimal diversity (357

**Funding:** The author(s) received no specific funding for this work.

**Competing interests:** The authors have declared that no competing interests exist.

living species) and over 50% are threatened with extinction [3, 4]. Understanding turtle biodiversity is difficult because they have low mitochondrial deoxyribonucleic acid (mtDNA) variation and low evolutionary change on a molecular level, which could be due to their slow maturity and unconstrained hybridization [5, 6].

The field of turtle genomics demonstrated major breakthroughs in 2013, beginning with the first three turtle genomes, and continues to progress to this day. In 2013, the entire genome of the western painted turtle (*Chrysemys picta bellii*) was sequenced, assembled, and examined through comparative genomic and phylogenetic analyses, highlighting that turtles are sister to archosaurs and evolve very slowly at the sequence level. The painted turtle possessed genes that allow them to tolerate reduced oxygen concentrations and freezing temperatures; analyses showed convergent evolution of tooth loss based on pseudogenization of those specific genes [7]. Furthermore, in the same year, the soft-shell turtle (*Pelodiscus sinensis*) and green sea turtle (*Chelonia mydas*) genomes were sequenced, leading to the discovery that a large expansion of olfactory genes that enhance their sense of smell [8]. The Pinta Island tortoise (*Chelonoidis abingdonii*) and Aldabra giant tortoise (*Aldabrachelys gigantea*) were sequenced in 2019 and provided insight into longevity and age-related diseases, specifically through duplications in age-related cell repair and cancer resistance genes. These tortoises also had mutations in DNA repair genes that carried through evolution [9]. Finally, whole duplications of toll-like receptor (TLR) genes were shown to trigger downstream immune response in the desert tortoise (*Gopherus agassizii*) [10]. With the advent of new, high throughput sequencing technologies and annotation tools, we have begun to uncover many discoveries about turtles.

While these automated genomic technologies are essential for our understandings about the turtle genome, they are prone to errors, resulting in incomplete and inaccurate annotations which may propagate across species [11]. A 2015 study falsely claimed that 17% of genes in the tardigrade (*Hypsibius dujardini*) draft genome were horizontally transferred from distant species [12]; however, this claim was refuted a year later when a more accurate assembly was generated after identifying and removing contaminants [13]. Also, because eukaryotic genomes comprise mostly of non-coding regions (i.e., introns), automated annotation is much more difficult compared to bacterial genomes. About 40–50% of eukaryotic genes are without functional annotation [14], and draft genomes are often highly fragmented with an unknown order [11]. Therefore, by using functional annotation, we minimize the risk of erroneously adopting inaccurate biological predictions to the gene models.

In this study, functional gene annotation was performed on two putative genes of the red-bellied short-neck turtle (*Emydura subglobosa*), a species whose genome has not been fully annotated. Until now, only 80 proteins have been computationally analyzed for *E. subglobosa* [15]. While genome sequencing, genome assembly, and automated annotations have been conducted on *E. subglobosa*, little is known about the evolutionary and molecular mysteries encoded in its genes due to the lack of functional gene annotation. Here, we aim to improve upon *ab initio* predictions and annotations provided by automated annotation tools, such as AUGUSTUS and Apollo. We focus our structural and functional analyses on mitochondrial protein-coding genes since previous work on this organism's genome have focused on the sex chromosomes [16] and Smad proteins [17]. While genomic annotation involves several types of data and methods, our strategy is exclusively based on homology and computational biology, using tools such as Basic Local Alignment Sequence Tool (BLAST), Constraint-based Multiple Alignment Tool (COBALT), InterPro, WoLF PSORT (Protein Subcellular Localization Prediction), Transmembrane Helices Hidden Markov Models (TMHMM), SWISS-MODEL, and STRING. Precise elucidation of the features encoded in the *E. subglobosa* genome gives insight into both the structure and function of the proteins and their relationship to

other eukaryotic species across evolutionary time, which can further advance conservation efforts and clinical medicine therapies.

# Materials and methods

## Genome sequencing and assembly

A prebuilt genome that was previously sequenced, assembled, and published on the NCBI Assembly database by the name of Emydura_subglobosa-1.0 (Accession: GCA_007922225.1) was utilized. The fully sequenced genome was approximately 2.6 Gb long with 43,399 scaffolds and 56,575 contigs [18, 19].

## *Ab initio* gene prediction

In this study, the AUGUSTUS gene prediction tool was used as a starting point for structural and functional analysis of the *E. subglobosa* genome. AUGUSTUS is based on a Hidden Markov Model (HMM) and performs *ab initio* gene prediction to define structural elements such as coding and non-coding regions, splice sites, and intergenic length distributions [20]. The AUGUSTUS prediction was initially evaluated through homology-based evidence and subsequently edited and/or validated through various genomic biology tools. This paper focused on the ML679947.1 scaffold of the *E. subglobosa* genome.

## Local fragment alignments

NCBI Protein Basic Local Alignment Sequence Tool (Protein BLAST or BLASTP) was used to compute alignments of the predicted query sequence against a protein database and calculate statistical significance of alignments based on similarities. This tool allowed the search for local alignment fragments with other species based on the maximal segment pair (MSP) and pairwise alignment algorithm [21]. After selecting a gene of interest on Apollo, the FASTA sequence of the predicted gene was inputted into the query search of standard Protein BLAST. The default settings of non-redundant protein sequences (nr) database and blastp algorithm were used. BLAST outputs included the description of top-hit homologs that shared common sub-sequences or fragments, percent identity for exact residue matches, query coverage, maximum high-scoring segment pair (HSP) score with no gaps, and E value showing statistical probability of chance HSP alignments [22]. Selection of the same isoform across the top 7 to 10 highest-matching species was done to avoid duplicating the same alignments. Only E values less than 0.1 and near zero were considered to ensure the low probability of the BLAST alignments occurring due to chance. This gave increased confidence that the alignment with the homolog was real and to deduce that the two gene copies were orthologs, not paralogs. Genes with a high percent identity and query coverage greater than 70% were also considered to find high conservation across different species. After selecting the top 7 to 10 best subject hits (same isoform) based on the pairwise alignment, the complete FASTA sequence was downloaded, rather than the aligned FASTA sequence, to obtain the entire sequences of all subjects in case the predicted model missed any information outside the aligned fragment. The predicted AUGUSTUS sequence was pasted to the top of the seqdump.txt file. These considerations led to increased confidence about the local alignment results from Protein BLAST, so that further analysis could be performed.

## Multiple sequence alignment

While Protein BLAST performs local alignment with no gaps, this algorithm does not guarantee optimal alignments due to potential evolutionary insertions and deletions (gaps).

Therefore, Constraint-based Multiple Alignment Tool (COBALT) was used to 1) detect other highly conserved, homologous protein sequences, 2) provide insight into protein function, structure, and evolution, and 3) identify sequencing errors and regions to be edited. Based on the cluster alignment (CLUSTAL) algorithm, COBALT uses pairwise alignment to calculate distance matrices, which are then continuously aligned based on a guide tree [23]. The seq-dump.txt FASTA file was uploaded to COBALT using default settings. COBALT outputted a structural schematic of the highly conserved (red) and variable (gray) regions of the protein sequence. At the bottom, exact protein sequences across conserved homologs were shown to determine which sequences were added, missing, or different in the query compared to the other validated sequences.

## Evidence-based genome editing and manual annotation

If the query showed distinct differences compared to the conserved protein homologs in other species, genome editing in Apollo was used to refine the AUGUSTUS prediction and generate a consensus model [24]. All edits were made from the perspective of the query compared to the homologous proteins. Gaps between exons or long gray exons were evidence of missing sequences or extra sequences in the coding region of the predicted gene model, respectively. For these missing sequences, the amino acid sequence that was present in the other homologs or found in the non-coding region nearby was pasted in the Search tab. "Blat protein" was selected. If a match was found, a new exon was made to encompass that specific amino acid sequence so that it was no longer missing. For long gray regions representing low conservation that were not present in other homologs, these extra exons were removed from the predicted protein by either deleting the entire exon or shortening it. If the predicted model was shorter than the other conserved sequences, this was evidence that the gene model was truncated and needed to be extended. New exons were created by 1) extending current exons, 2) splitting the current exon and dragging the new one to the appropriate region, or 3) merging two different genes. Short missing or extra sequences found in the middle of an exon were unedited since they could potentially be due to evolutionary insertions or deletions. All edits were carefully completed to maintain the open reading frame for proper transcription. The gene model was iteratively refined using the three tools—BLAST, COBALT, and Apollo—to generate a new consensus model that was comparable to the conserved genomes based on homology.

## Functional analysis

To determine the function(s) of the protein, InterPro, AmiGO 2, and STRING v. 11.0 were used. The purpose of InterPro was to determine protein function based on similar sequence motifs and domains [25]. Because domains are functional parts of a protein and are highly conserved, inferring function through domain searches allowed us to classify the protein of interest into families. The FASTA sequence of the consensus model was pasted into the "Search by sequence" box and default settings were used. The protein family membership was noted. InterPro allows for the visualization of the different families, domains, superfamilies, and predictions that stretched along the length of the gene at varying lengths. Gene ontology (GO) terms for biological processes, molecular functions, and cellular components were also outputted at the domain level and recorded. In addition, AmiGO 2 was used to complement InterPro's GO terms by generating top hits of GO terms associated with the protein itself, rather than at the domain level [26]. The homologous gene name associated with the protein was determined and the ontology graph under the Graph Views tab was evaluated, which gave insight into the larger biological process that the protein was involved in.

For further functional analysis, STRING v. 11.0 was used to predict direct and indirect protein-protein interactions [27]. Under the "Protein by sequence" sidebar, the sequence was pasted into the "Amino Acid Sequence" box and the organism was "*Homo sapiens.*" Humans were used as the organism to investigate conservation of interacting proteins and determine whether these interaction networks were conserved in turtles. The outputted network from STRING highlighted several key interactions and functional partners. Under the "Viewers" tab, gene cooccurrence was examined: dark red squares corresponding to gene families whose occurrence patterns showed strong homology across species, light red squares corresponding to gene families whose occurrence patterns showed weak homology across species, and missing squares corresponding to gene families whose occurrence patterns showed no direct homologies across species.

## Subcellular localization

To identify the subcellular localization, different complementary tools were utilized, including WoLF PSORT (Protein Subcellular Localization Prediction), Transmembrane Helices Hidden Markov Models (TMHMM) Server v. 2.0, SignalP-5.0 Server, Phobius, and TargetP-2.0 Server. WoLF PSORT is a tool that predicts protein localization sites based on the primary amino acid composition [28]. "Animal" was selected as the organism type, "From Text Area" was kept as default, and the FASTA sequence of the consensus model was pasted into WoLF PSORT. The highest scoring matches based on percent identity were noted which predicted where the protein was localized. TMHMM Server v. 2.0 further complemented results from InterPro and WoLF PSORT by prediction of transmembrane helices (TMHs). TMHMM uses the probabilistic Hidden Markov Model (HMM) developed from observed sequences of proteins with known functions [29]. The FASTA sequence was pasted and utilized default settings. The top schematic summarized discrete regions within the protein. Probabilities greater than 0.75 were considered significant in indicating segments of the protein that lie inside, outside, or within the membrane. Additionally, SignalP-5.0 Server was used to predict the signal peptide cleavage site that targets the protein to its correct location [30]. The FASTA sequence of the protein was pasted and the default settings of "Eukarya" organism group and "long output" format were used. Probabilities greater than 0.5 suggested that the gene encodes a signal peptide; probabilities greater than 0.75 gave greater confidence regarding the presence of a cleavage site. To confirm the outputs of TMHMM and SignalP, Phobius was used to predict TMHs and signal peptides [31]. Again, the FASTA sequence was pasted with default settings, and probabilities greater than 0.75 were considered significant. Finally, TargetP-2.0 was used to identify N-terminal presequences, signal peptides, and transit/targeting peptides [32]. The FASTA sequence was pasted with default settings of "Non-Plant" organism group and "long output" format. Probabilities greater than 0.5 suggested that the gene encodes an N-terminal pre-sequence, signal peptide, or transit peptide; probabilities greater than 0.75 gave greater confidence. Overall, these tools provided insight into where the protein localizes within the cell.

## Protein modeling and structure

To complement the TMHMM results, SWISS-MODEL was also used to generate a three-dimensional (3D) protein structure based on homology [33]. The FASTA sequence was pasted into the "Target Sequence" and the model was built. The SWISS-MODEL output consisted of information regarding oligo-state, ligands, global (GMQE) and local (LMQE) model quality estimations, model-template alignments corresponding to secondary structures, comparison with non-redundant set of protein database (PDB) structures, and an overall 3D protein

structure model. The model with the GMQE closest to 1 was examined, and results were confirmed with TMHMM regarding transmembrane helices.

To validate the model, the LMQE, comparison with non-redundant set of PDB structures, Ramachandran Plot, and MolProbity results were examined. The LMQE shows pair residue estimates, and scores <0.6 were considered low-quality. For the graphical comparison with non-redundant set of PDB structures, the dark gray represented the QMEAN score for experimental structures of similar size; a good model was considered to fall within the gray range. The Ramachandran Plot under "Structure Assessment" shows the probability of a residue having a specific orientation. The model was validated if it had a Ramachandran Favoured value close to 100%, Ramachandran Outliers value close to 0%, and MolProbity Score close to 0. The final tool used to validate the homology structure was PSIPRED 4.0, in which the protein sequence was pasted and submitted using default settings. PSIPRED predicts secondary structure, which also complemented TMHMM results [34].

## Phylogenetic tree analysis

The phylogenetic tree functions of BLAST and COBALT were used to scrutinize the evolution of the protein across various species and visualize the graphical relation between genomic sequences in tree format. BLAST and COBALT use pairwise alignment and multiple sequence alignment, respectively, to construct the tree [21, 23]. More similar sequences branched closer to the right, and less similar sequences branched closer to the left. The phylogenetic tree was built and expanded out with organism sets by including disparate species (i.e., species with lower percent identities in the BLAST results) into the seqdump.txt file. This was to confirm the relative similarities between homologs versus distant species and serve as a comparison group when examining the relative position of *E. subglobosa* in the tree. BLAST and COBALT gave the ability to determine how similar the other homologs aligned to the predicted sequence. Table 1 lists all tools used in the methods.

## Results

### Mitochondrial cholesterol side-chain cleavage enzyme (CYP11A1)

The first gene model analyzed was the g19.t1 gene located within the ML679947.1 scaffold. The AUGUSTUS gene prediction contained 520 residues and 9 exons (Fig 1A, S1 Table). Local pairwise alignment on BLAST showed a high query coverage of 98–100%, high percent identity ranging from 83–87%, and low E value of 0, suggesting a very low probability that alignments were obtained by chance (Table 2). We can be confident that this is a real alignment and that these gene copies are orthologs and not paralogs. The graphical distribution of top 10 BLAST hits showed a query coverage across the entire gene at high conservation (Fig 1B). Through the COBALT multiple sequence alignment tool, the peptide sequence was aligned and had high conservation and similarity across other homologous proteins (Fig 1C). The predicted gene model was missing 3 residues from positions 488–490 when comparing the query sequence to the homologous sequences (Fig 1C). The two exons in Query_10009 and Query_10011 were not edited because those extra exons were specific to that sequence and not found in any others (Fig 1C). We could not find the sequence of the missing region nor the nearby region in the gene model, suggesting that a deletion occurred during the evolution of *E. subglobosa*. Therefore, we concluded that AUGUSTUS predicted the most refined consensus model.

After validation of the gene model, we analyzed the predicted function of this peptide sequence through InterPro and STRING. InterPro predicted that the protein was a mitochondrial cholesterol side-chain cleavage enzyme (Fig 2A), which concurred with BLAST (Fig 1B).

**Table 1. Summary table of 15 computational biology tools used to study the conservation, structure, function, localization, and phylogeny of genes within the *E. subglobosa* scaffold.**

| Name | Type | Description | URL | Access Date | Citation | Reference |
|---|---|---|---|---|---|---|
| **AmiGO 2** | Database | Gene ontology | http://amigo.geneontology.org/amigo | April 2021 | (AmiGO, RRID:SCR_002143) | [26] |
| **Apollo v. 2.6.2** | Software, algorithm | Environment for genome annotations, editing, and refinements to generate a consensus model | http://164.67.110.195/annotator/index | April 2021 | (Apollo, RRID:SCR_001936) | [24] |
| **AUGUSTUS** | Software, algorithm | Gene prediction | http://bioinf.uni-greifswald.de/augustus/ | April 2021 | (Augustus, RRID:SCR_008417) | [20] |
| **NCBI Protein BLAST or BLASTP** | Software, algorithm | Compares protein model to database of protein homologs in biologically similar species; phylogenetic tree analysis | https://blast.ncbi.nlm.nih.gov/Blast.cgi | April 2021 | (BLASTP, RRID:SCR_001010) | [21, 22] |
| **NCBI COBALT** | Software, algorithm | Multiple sequence alignment; compares predictive regions of the genome with other homologous sequences; phylogenetic tree analysis | https://www.ncbi.nlm.nih.gov/tools/cobalt/re_cobalt.cgi | April 2021 | (Cobalt: Constraint-based Multiple Alignment Tool, RRID:SCR_004152) | [23] |
| **InterPro** | Software, algorithm | Protein function based on domains and classification into families; gene ontology | http://www.ebi.ac.uk/interpro/ | April 2021 | (InterPro, RRID:SCR_006695) | [25] |
| **NCBI Assembly** | Database | Provides information on assembled genomes of various organisms | https://www.ncbi.nlm.nih.gov/assembly/ | April 2021 | (NCBI Assembly Archive Viewer, RRID:SCR_012917) | [18, 19] |
| **Phobius** | Software, algorithm | Predicts whether the protein has transmembrane helices (TMHs) and signal peptides | https://phobius.sbc.su.se/ | May 2021 | (Phobius, RRID:SCR_0156643) | [31] |
| **PSIPRED 4.0** | Software, algorithm | Secondary structure prediction | http://bioinf.cs.ucl.ac.uk/psipred/ | May 2021 | (PSIPRED, RRID:SCR_010246) | [34] |
| **SignalP-5.0 Server** | Software, algorithm | Signal peptide cleavage site prediction for organelle targeting | https://services.healthtech.dtu.dk/service.php?SignalP | May 2021 | (SignalP, RRID:SCR_015644) | [30] |
| **STRING v. 11.0** | Database | Looks at conservation of interacting genes across other species; protein-protein interaction networks; functional enrichment analysis | https://string-db.org | April 2021 | (STRING, RRID:SCR_005223) | [27] |
| **SWISS-MODEL** | Software, algorithm | Secondary structure prediction; homology modeling of three-dimensional protein structure | https://swissmodel.expasy.org | April 2021 | (SWISS-MODEL, RRID: SCR_018123) | [33] |
| **TargetP-2.0 Server** | Software, algorithm | Prediction of N-terminal presequences, signal peptides, and transit peptides | https://services.healthtech.dtu.dk/service.php?TargetP | May 2021 | (TargetP, RRID:SCR_019022) | [32] |
| **TMHMM Server v. 2.0** | Software, algorithm | Predicts whether the protein has transmembrane helices (TMHs) | http://www.cbs.dtu.dk/services/TMHMM/ | April 2021 | (TMHMM Server, RRID: SCR_014935) | [29] |
| **WoLF PSORT** | Software, algorithm | Prediction of protein localization sites based on primary amino acid composition | https://wolfpsort.hgc.jp | April 2021 | (WoLF PSORT, RRID: SCR002472) | [28] |

The protein belonged to three families: cytochrome P450 (IPR001128), CYP11A1 (IPR033283), and cytochrome P450 E-class group I (IPR002401, Fig 2A). The predicted homologous superfamily was cytochrome P450 (IPR036396), and there was also a conserved site from residues 457–466 (IPR036396, Fig 2A). The protein was predicted to be involved in the biological process of C21-steroid hormone biosynthetic processes (GO:0006700, Fig 2B). Its predicted molecular functions included oxidoreductase activity (GO:0016705), heme binding (GO:0020037), monooxygenase activity (GO:0004497), iron ion binding (GO:0005506), and cholesterol monooxygenase (side-chain-cleaving) activity (GO:0008386, Fig 2B). Its predicted cellular component was the mitochondrion (GO:0005739, Fig 2B).

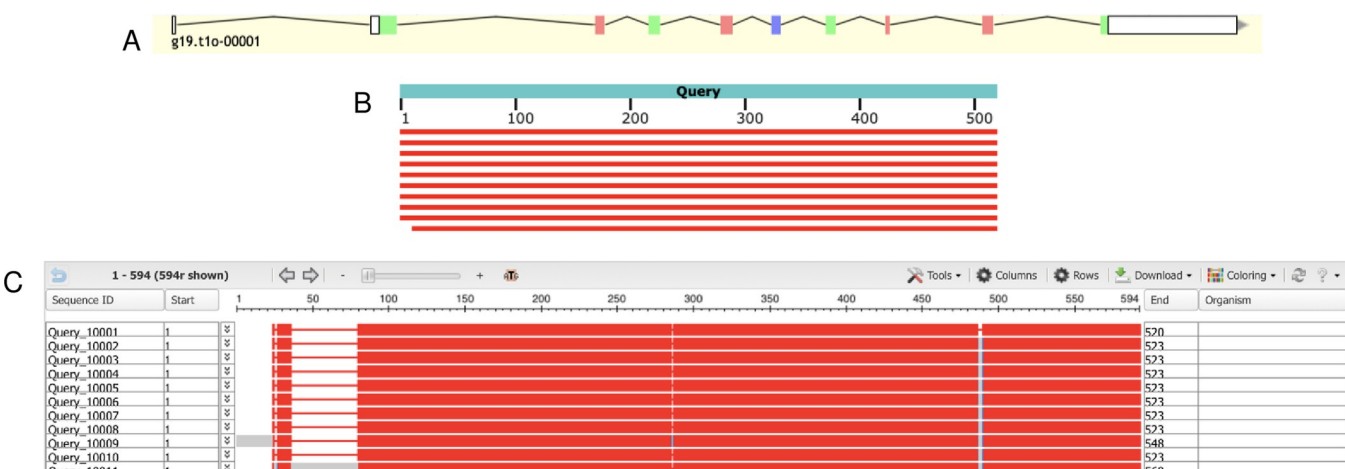

**Fig 1. Homology-based genome annotation of the cholesterol side-chain cleavage enzyme.** (A) Apollo gene editor view and AUGUSTUS track of the g19.t1 gene located within the ML679947.1 scaffold. (B) Graphical representation of query coverage across the top 10 BLAST hits on 10 subject sequences. Red means high conservation. (C) COBALT multiple sequence alignment demonstrating high conservation (red) across the homologs. Low conservation is colored gray. Exons (thick lines) and introns (thin lines) are shown. Query sequence is the top, while the subjects are below.

To further analyze function, we also determined protein-protein interactions through the STRING database. The protein was predicted to be CYP11A1, which plays a role in catalyzing the side-chain cleavage reaction of cholesterol in the mitochondria (Fig 2C and 2D); this agreed with the InterPro protein family results (Fig 2A). The protein's predicted functional partners included FDX1 (a mitochondrial adrenodoxin that synthesizes thyroid hormones), STAR (a mitochondrial steroidogenic acute regulatory protein), HSD3B1/2 (dehydrogenase enzymes), and several others (Fig 2C and 2D). These network links suggest co-evolution of

**Table 2. BLAST output of the g19.t1 sequence.** Top hits predicted a mitochondrial cholesterol side-chain cleavage enzyme with a high query coverage, high percent identity, and low E value.

| Description | Scientific Name | Max Score | Query Cover | E Value | Percent Identity (%) | Accession |
|---|---|---|---|---|---|---|
| cholesterol side-chain cleavage enzyme, mitochondrial | *Terrapene carolina triunguis* | 935 | 100% | 0 | 87 | XP_024070917.1 |
| cholesterol side-chain cleavage enzyme, mitochondrial | *Platysternon megacephalum* | 931 | 100% | 0 | 86.62 | TFK04060.1 |
| cholesterol side-chain cleavage enzyme, mitochondrial isoform X1 | *Chrysemys picta bellii* | 931 | 100% | 0 | 86.42 | XP_023959792.1 |
| cholesterol side-chain cleavage enzyme, mitochondrial isoform X1 | *Trachemys scripta elegans* | 927 | 100% | 0 | 86.23 | XP_034640650.1 |
| cholesterol side-chain cleavage enzyme, mitochondrial | *Chelonia mydas* | 927 | 100% | 0 | 85.66 | XP_007055541.1 |
| cholesterol side-chain cleavage enzyme, mitochondrial | *Mauremys reevesii* | 917 | 100% | 0 | 85.28 | XP_039346198.1 |
| cholesterol side-chain cleavage enzyme, mitochondrial | *Chelonoidis abingdonii* | 915 | 100% | 0 | 85.09 | XP_032629904.1 |
| cholesterol side-chain cleavage enzyme, mitochondrial isoform X1 | *Pelodiscus sinensis* | 905 | 100% | 0 | 83.59 | XP_025039409.1 |
| cholesterol side-chain cleavage enzyme, mitochondrial | *Gopherus evgoodei* | 893 | 100% | 0 | 82.41 | XP_030401535.1 |
| cholesterol side-chain cleavage enzyme, mitochondrial | *Dermochelys coriacea* | 885 | 98% | 0 | 83.24 | XP_038275216.1 |

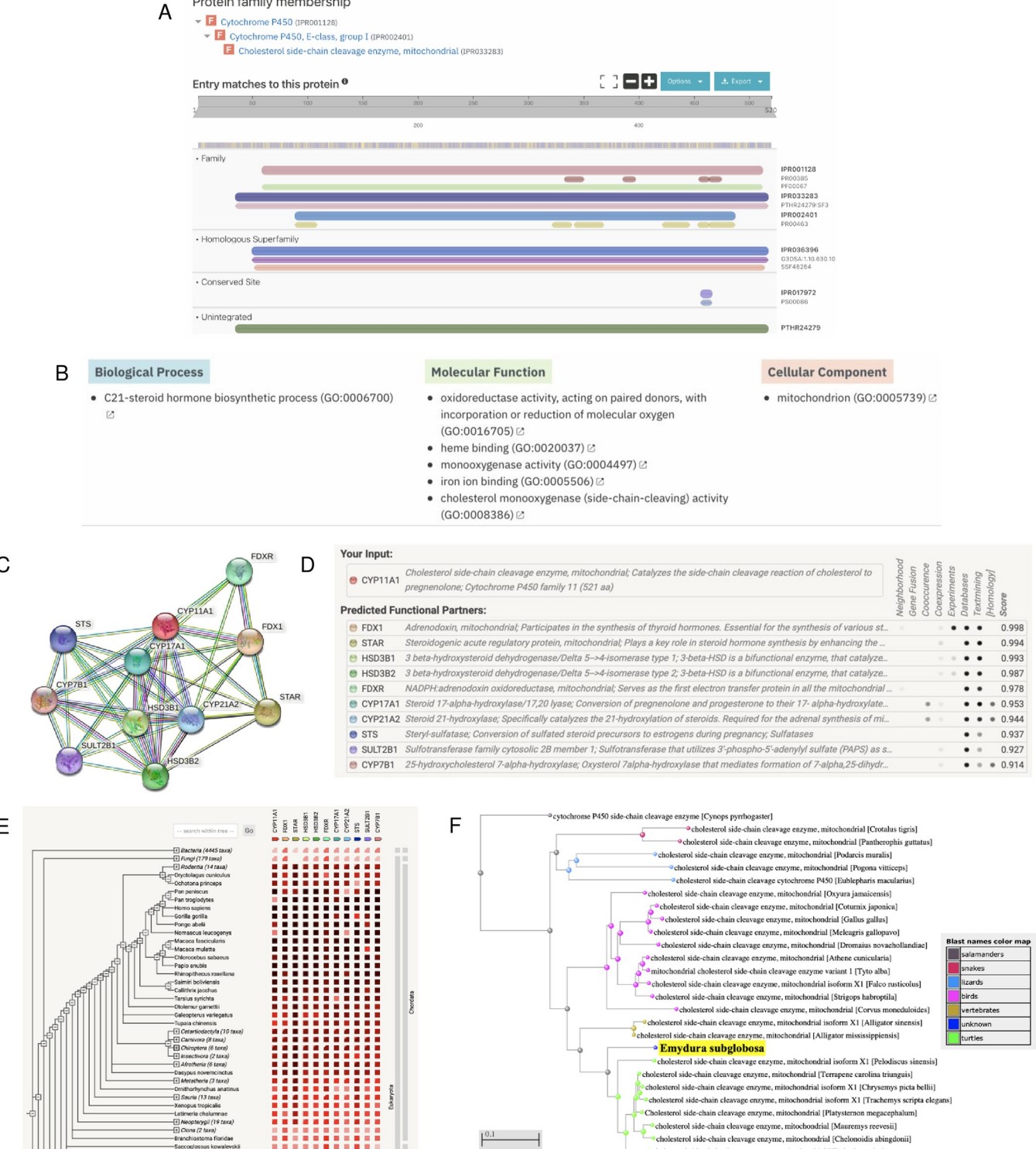

**Fig 2. Functional analysis of the cholesterol side-chain cleavage enzyme.** (A) InterPro functional analysis of the enzyme. (B) GO terms for the enzyme outputted by InterPro. (C) STRING network of predicted protein-protein interactions in *H. sapiens*. (D) List of functional partners predicted by STRING corresponding to C. (E) Gene co-occurrence of the protein. (F) BLAST phylogenetic tree built based on pairwise alignment.

non-homologous proteins since they are functionally related and could warrant further analysis. The interaction between CYP11A1 and FDX1 was determined experimentally, while the interaction between CYP11A1 and CYP17A1, CYP21A2, and CYP7B1 were determined through homology (Fig 2D). Further experimental research can be done to determine whether they act in a complex or directly bind each other. Additionally, the gene co-occurrence predicted that CYP11A1 interacts with CYP17A1 and CYP21A2, which suggests that these gene families have strong homology across the Chordata phylum (Fig 2E). Finally, based on phylogenetic tree analysis of the mitochondrial cholesterol side-chain cleavage enzyme, *E. subglobosa* is a sister group to other turtle species in the phylogenetic tree, suggesting that the gene copies are orthologs rather than paralogs (Fig 2F and S1 Fig).

We also wanted to determine the subcellular localization of the protein based on any TMHs, signal sequences/peptides, and transit peptides. In WoLF PSORT, a large majority of the highest matches predicted a mitochondrial localization, suggesting that this protein is likely located in the mitochondria (Fig 3A). There were no TMHs predicted, so the protein presumably does not localize on or near the plasma membrane (Fig 3B). The protein did not have any predicted signal peptides (Fig 3C). Phobius predictions also agreed with both TMHMM and SignalP (Fig 3D). Despite having no signal peptide to target the protein to the mitochondria, there was statistically significant evidence of a mitochondrial transfer peptide at positions 36–37 (P = 0.8743, Fig 3E). Based on the localization analysis, several pieces of evidence agree that the protein localizes to the mitochondria.

After characterizing the function and localization of the enzyme, we also wanted to understand its structure since protein structure heavily influences protein function. SWISS-MODEL predicted a monomeric structure that was highly conserved based on homology (Fig 4A and 4B). To validate the model, we examined the global and local quality estimates. The GMQE was 0.74, QMEAN was -1.77, and sequence identity was 54.04% (Fig 4B). Additionally, the local quality estimates had high probabilities in a majority of the regions; areas that dipped below 0.6 may have had poor resolution or were less conserved (Fig 4C). Comparing the model to a non-redundant set of PDB structures, the QMEAN was relatively near the average QMEAN scores for proteins of similar size (Fig 4D). The Ramachandran Plot also showed high favorability of the residues having a specific orientation, as shown by a majority of dots (95.91%) located in the dark green regions (Fig 4E, Table 3). The MolProbity Score was near 0; the clash score, outliers, deviations, and bad bonds/angles were low (Table 3). Based on PSIPRED, there were both alpha-helixes and beta strands in the secondary structure, which was consistent with SWISS-MODEL (Fig 4F). Together, these data suggest a high-quality structure that could be used to model the mitochondrial cholesterol side-chain cleavage enzyme.

## Mitochondrial methylmalonyl-CoA epimerase (MCEE)

The second gene model analyzed was the g112.t1 gene located within the ML679947.1 scaffold. The AUGUSTUS gene prediction contained 122 residues and 3 exons (Fig 5A, S1 Table). Local pairwise alignment on BLAST showed a high query coverage of 98%, moderate percent identity ranging from 55–59%, and low E values, suggesting a very low probability of alignments to be obtained by chance (Table 4). We can be confident that this alignment did not occur by chance and that these gene copies are orthologs and not paralogs. The graphical distribution of top 7 BLAST hits also aligns with the aforementioned results, showing a query coverage across the entire gene but relatively moderate conservation (Fig 5B). Through the COBALT multiple sequence alignment tool, the peptide sequence was aligned and had high conservation and similarity across other homologous proteins in certain areas (Fig 5C). When

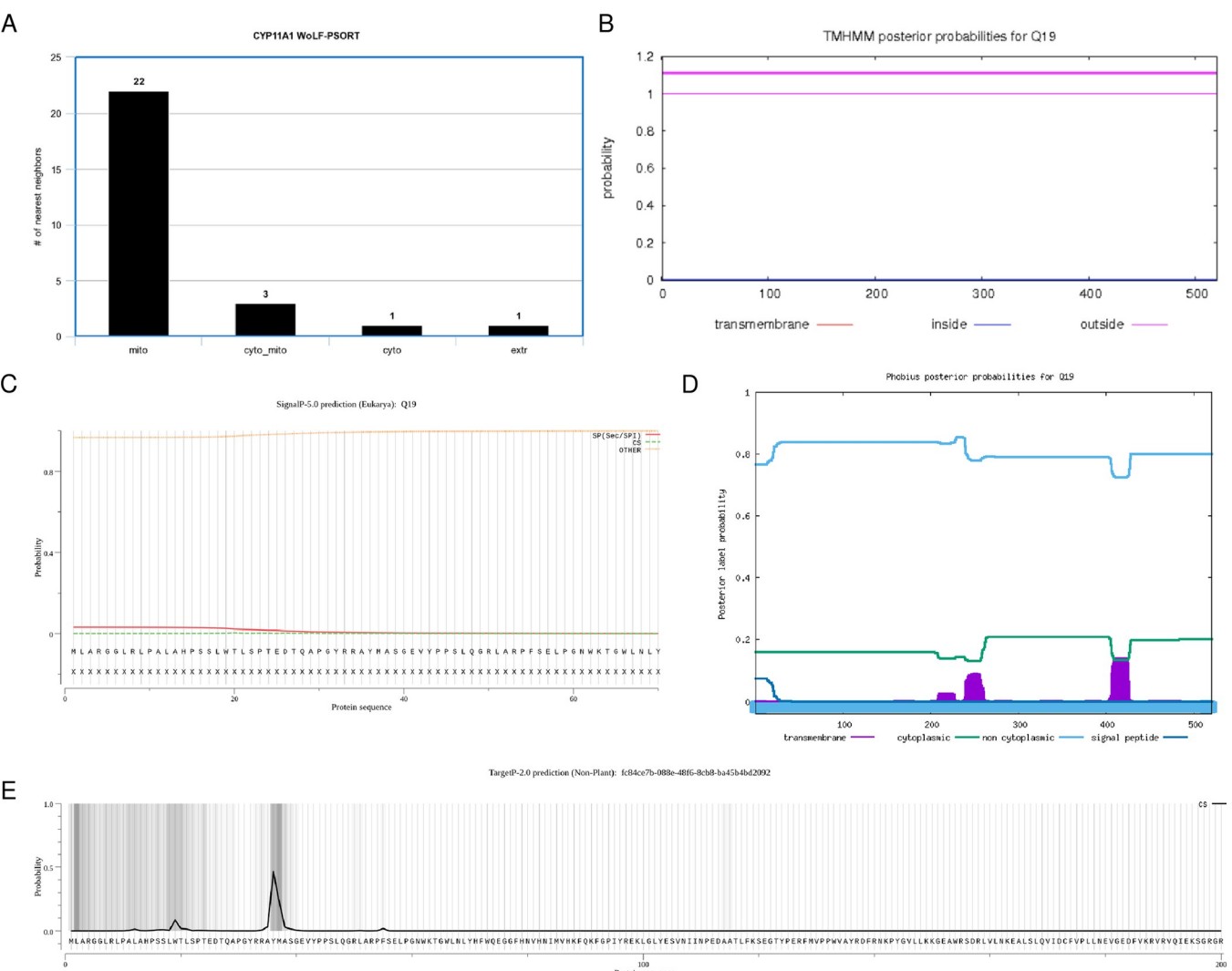

**Fig 3. Subcellular localization of the cholesterol side-chain cleavage enzyme.** (A) Bar chart displaying WoLF PSORT prediction of the protein's localization sites based on 32 nearest neighbors. Mito, mitochondria; cyto_mito, cytoplasm and mitochondria; cyto, cytoplasm; extr, extracellular. (B) TMHMM prediction of TMHs. X-axis represents the amino acid number, and y-axis represents the probability that the amino acid is located within, outside, or inside the membrane. Probabilities >0.75 are significant. (C) SignalP analysis of signal sequences existing in the amino acid sequence of the polypeptide. (D) Phobius predictions of TMHs and signal peptides. X-axis represents the amino acid number, and y-axis represents the probability that the amino acid is transmembrane, cytoplasmic, non-cytoplasmic, and/or a signal peptide. Probabilities >0.75 are significant. (E) TargetP-2.0 prediction of N-terminal pre-sequences, signal peptides, and transit peptides.

comparing the query sequence to the homologous sequences, the predicted gene model was missing 54 residues from positions 82–135 and had 1 extra residue at position 20 (Fig 5C). For the missing sequences, we extended the exon to encompass the missing residues in the coding region of the gene; for the extra residue, we shortened the exon by one amino acid since it was on the end of the exon. Therefore, the most refined consensus model contained 174 residues and 4 exons (Fig 5A, S1 Table). BLAST results after gene editing revealed a higher query coverage of 100% and higher percent identity ranging from 90–94% (Table 5), and COBALT alignment was highly conserved across the gene (Fig 5C).

After validation of the gene model, we analyzed the predicted function of this peptide sequence through InterPro, AmiGO 2, and STRING. InterPro predicted that the protein was a

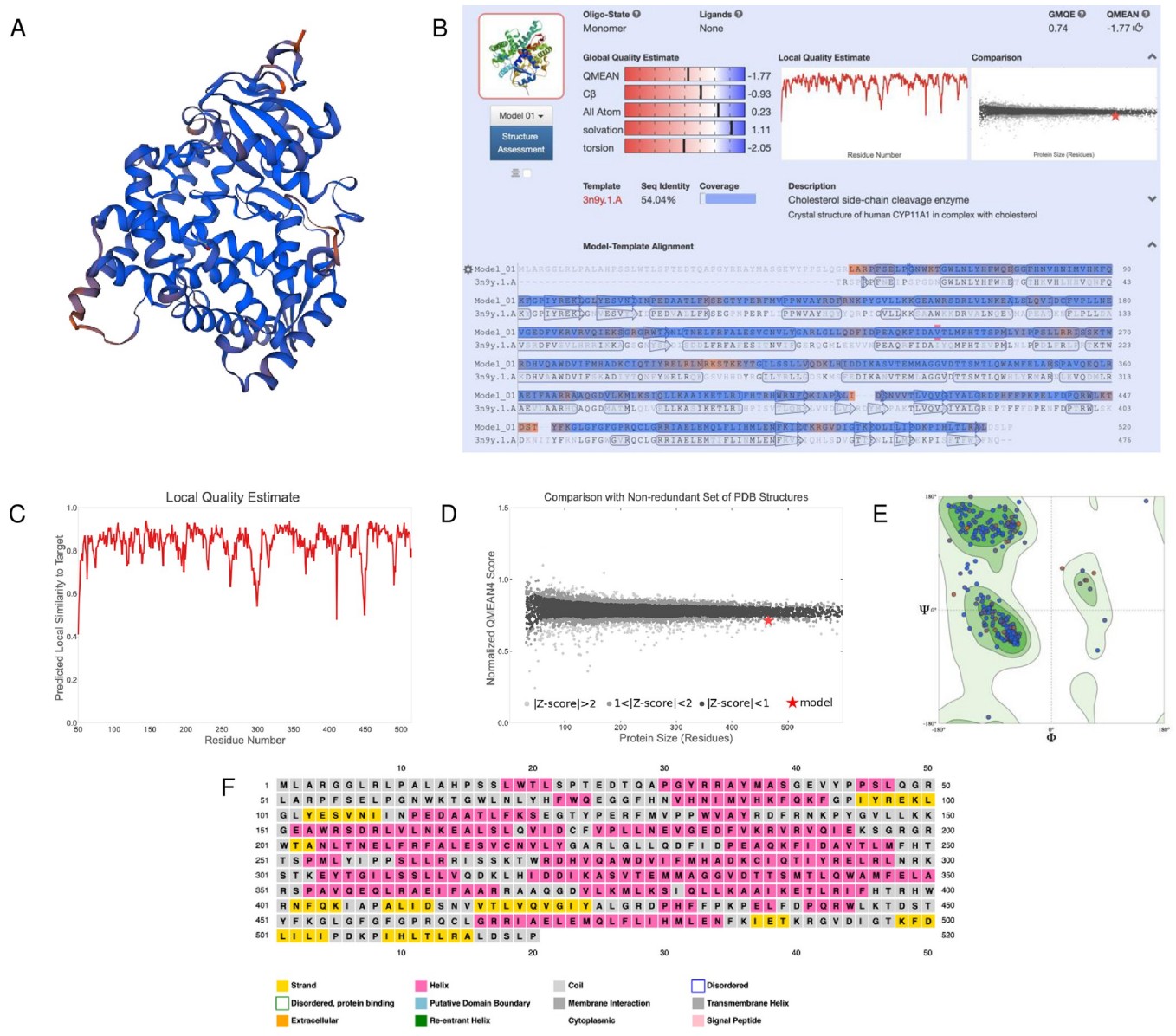

**Fig 4. Homology modeling and structural predictions of the mitochondrial cholesterol side-chain cleavage enzyme.** (A) Three-dimensional homology model built by SWISS-MODEL. Blue regions are highly conserved, while orange regions are less conserved. (B) Oligo-state, ligands, global quality estimates, template, sequence identity, and coverage outputted by SWISS-MODEL. (C) Local quality estimate showing pair residue estimates. Similarities >0.6 are high-quality models. (D) Comparison with non-redundant set of PDB structures showing QMEAN scores for experimental structures that have been deposited of similar size. The red star is our model. (E) Ramachandran plot showing the probability of a residue having a specific orientation. Dots in the dark green regions represent high probability and a high-quality model. (F) Secondary structure prediction through PSIPRED.

methylmalonyl-CoA epimerase (Fig 6A), which concurred with BLAST (Fig 5B). The protein belonged to the methylmalonyl-CoA epimerase family (IPR017515) with metal and substrate binding sites and dimer interfaces (Fig 6A). The protein was also predicted to be a part of the VOC domain (IPR037523) and glyoxalase/bleomycin resistance protein/dihydroxybiphenyl dioxygenase homologous superfamily (IPR029068, Fig 6A). There were also predictions of a signal peptide and non-cytoplasmic domains (Fig 6A). No GO terms were outputted on Inter-Pro; however, two GO terms outputted by AmiGO 2 were the biological process L-

**Table 3. MolProbity results to validate the SWISS-MODEL prediction for CYP11A1.** A MolProbity Score close to 0 represents the resolution that one would see a structure of this quality. Clash score represents overlapping residues; a lower value is favored. Outliers represent values that extend outside the standard deviation; low values are also favored. Low values for bad bonds and angles are also favored.

| | |
|---|---|
| **MolProbity Score** | 1.44 |
| **Clash Score** | 2.99 |
| **Ramachandran Favoured** | 95.91% |
| **Ramachandran Outliers** | 0.86% |
| **C-Beta Deviations** | 6 |
| **Bad Bonds** | 0/3908 |
| **Bad Angles** | 46/5287 |

methylmalonyl-CoA metabolic (GO:0046491) and the molecular function methylmalonyl-CoA epimerase (GO:0004493, S2 Fig).

To further analyze function, we also determined protein-protein interactions through the STRING database. The protein was predicted to be MCEE, which belongs to the glyoxalase I family (Fig 6B and 6C); this agreed with the InterPro protein family results (Fig 6A). The enzyme's predicted functional partners included MUT (methylmalonyl-CoA mutase involved in degrading amino acids), MMAA (hydrolyzes GTP), ECHDC1 (decarboxylase), and several others (Fig 6B and 6C). These network links suggest the co-evolution of the non-homologous proteins since they are functionally related and could warrant additional analysis. Further research can also be done to determine whether they act in a complex or directly bind each other. Additionally, the gene co-occurrence suggests that these gene families have strong homology across the Chordata phylum (Fig 6D). Finally, based on phylogenetic tree analysis of the methylmalonyl-CoA epimerase, *E. subglobosa* shared the same root as other turtle species in the phylogenetic tree, suggesting that these gene copies came from a common ancestor (Fig 6E and S3 Fig).

We also wanted to analyze the subcellular localization of the protein based on any TMHs, signal sequences/peptides, and transit peptides. In WoLF PSORT, a majority of the highest matches predicted the mitochondria, suggesting that this protein is likely located in the mitochondria (Fig 7A). There were no TMHs predicted and only non-cytoplasmic domains, so the protein does not localize to the plasma membrane (Fig 7B). The protein was predicted to not have a signal peptide sequence in both SignalP (Fig 7C) and TargetP (Fig 7E) but had very high probability for a mitochondrial transfer peptide (P = 0.9816, not shown). However, Phobius predictions did not agree with SignalP and showed a signal peptide from residues 1–20, so further validation and ribonucleic acid sequencing (RNA-seq) may be needed to gain confidence in whether the protein has a signal peptide (Fig 7D). Based on our analyses, the protein likely localizes to the mitochondria.

After identifying the function and localization of the enzyme, structure was investigated. The SWISS-MODEL predicted a homo-dimeric structure that was highly conserved based on homology (Fig 8A and 8B). The GMQE was 0.59, QMEAN was 1.44, and sequence identity was 81.06% (Fig 8B). Additionally, the local quality estimates had high probabilities in most regions; areas that dipped below 0.6 may have had poor resolution or were less conserved (Fig 8C). Notably, the homodimers were also extremely similar. Comparing the model to a non-redundant set of PDB structures, the QMEAN was within the average QMEAN scores for proteins of similar size (Fig 8D). The Ramachandran Plot also showed high favorability of the residues having a specific orientation, as shown by most dots (97.31%) located in the dark green regions (Fig 8E, Table 6). The MolProbity Score was near 0; the clash score, outliers,

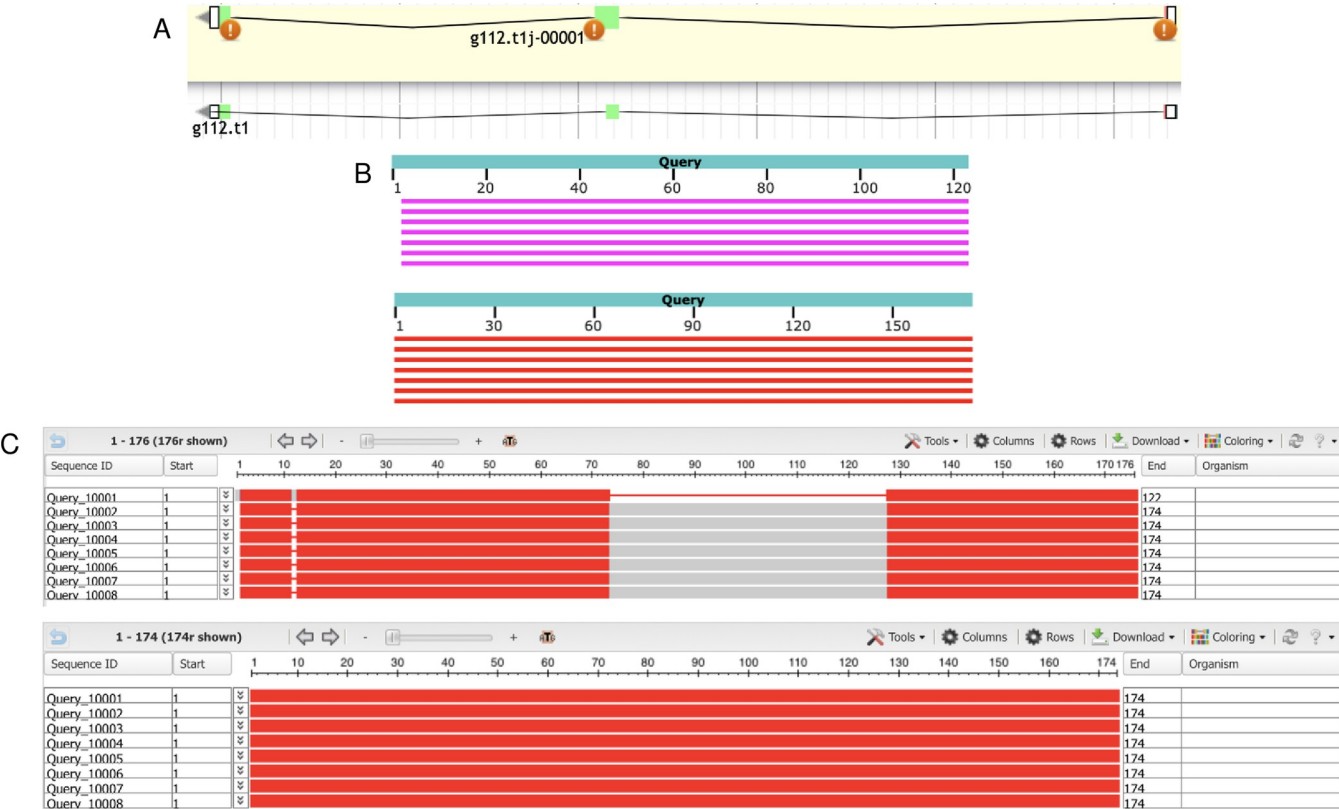

**Fig 5. Homology-based genome annotation of the methylmalonyl-CoA epimerase (MCEE) enzyme.** (A) Apollo gene editor view and AUGUSTUS track of the g112.t1 gene located within the ML679947.1 scaffold. Bottom: initial *ab initio* prediction. Top: consensus gene model. (B) Graphical representation of query coverage across the top 7 BLAST hits on 7 subject sequences before (top) and after (bottom) genome editing. Red means high conservation, and magenta means moderate conservation. (C) COBALT multiple sequence alignment before (top) and after (bottom) genome editing, demonstrating high conservation (red) across the homologs. Low conservation is colored gray. Exons (thick lines) and introns (thin lines) are shown. Query sequence is the top, while the subjects are below.

deviations, and bad bonds/angles were low (Table 6). PSIPRED predicted both alpha-helixes and beta-strands in the secondary structure, which was consistent with SWISS-MODEL (Fig 8F). Together, these data suggest a high-quality structure that could be used to model the methylmalonyl-CoA epimerase enzyme.

**Table 4. BLAST output of the g112.t1 sequence before genome editing.** Top hits predicted mitochondrial methylmalonyl-CoA epimerase with a high query coverage, moderate percent identity, and low E value.

| Description | Scientific Name | Max Score | Query Cover | E Value | Percent Identity (%) | Accession |
|---|---|---|---|---|---|---|
| methylmalonyl-CoA epimerase, mitochondrial isoform X1 | *Chrysemys picta bellii* | 180 | 98% | 3.00E-55 | 58.62 | XP_005285039.1 |
| methylmalonyl-CoA epimerase, mitochondrial isoform X1 | *Terrapene carolina triunguis* | 178 | 98% | 1.00E-54 | 58.05 | XP_024048902.1 |
| methylmalonyl-CoA epimerase, mitochondrial isoform X1 | *Dermochelys coriacea* | 177 | 98% | 4.00E-54 | 58.05 | XP_038274304.1 |
| methylmalonyl-CoA epimerase, mitochondrial | *Chelonia mydas* | 177 | 98% | 6.00E-54 | 58.05 | XP_007053822.1 |
| methylmalonyl-CoA epimerase, mitochondrial isoform X1 | *Trachemys scripta elegans* | 176 | 98% | 1.00E-53 | 58.05 | XP_034640814.1 |
| methylmalonyl-CoA epimerase, mitochondrial isoform X1 | *Mauremys reevesii* | 173 | 98% | 2.00E-52 | 56.32 | XP_039349013.1 |
| methylmalonyl-CoA epimerase, mitochondrial isoform X1 | *Chelonoidis abingdonii* | 168 | 98% | 1.00E-50 | 54.6 | XP_032631950.1 |

**Table 5. BLAST output of the g112.t1 sequence after genome editing.** Top hits also predicted mitochondrial methylmalonyl-CoA epimerase with a higher query coverage, higher percent identity, and lower E value.

| Description | Scientific Name | Max Score | Query Cover | E Value | Percent Identity (%) | Accession |
|---|---|---|---|---|---|---|
| methylmalonyl-CoA epimerase, mitochondrial isoform X1 | *Terrapene carolina triunguis* | 340 | 100% | 9.00E-118 | 94.25 | XP_024048902.1 |
| methylmalonyl-CoA epimerase, mitochondrial isoform X1 | *Chrysemys picta bellii* | 340 | 100% | 2.00E-117 | 94.25 | XP_005285039.1 |
| methylmalonyl-CoA epimerase, mitochondrial | *Chelonia mydas* | 340 | 100% | 2.00E-117 | 94.25 | XP_007053822.1 |
| methylmalonyl-CoA epimerase, mitochondrial isoform X1 | *Dermochelys coriacea* | 337 | 100% | 2.00E-116 | 93.68 | XP_038274304.1 |
| methylmalonyl-CoA epimerase, mitochondrial isoform X1 | *Trachemys scripta elegans* | 336 | 100% | 6.00E-116 | 93.68 | XP_034640814.1 |
| methylmalonyl-CoA epimerase, mitochondrial isoform X1 | *Mauremys reevesii* | 336 | 100% | 6.00E-116 | 93.1 | XP_039349013.1 |
| methylmalonyl-CoA epimerase, mitochondrial isoform X1 | *Chelonoidis abingdonii* | 329 | 100% | 2.00E-113 | 90.23 | XP_032631950.1 |

## Discussion

Our study manually annotated and improved upon two predicted gene models in the ML679947.1 scaffold of the *E. subglobosa* genome. There is strong evidence that these genes encode a cholesterol side-chain cleavage enzyme and methylmalonyl-CoA epimerase—both of which are localized in the mitochondria and are linked by the TCA cycle. Initial BLAST pairwise alignments and COBALT multiple sequence alignments showed these genes were highly conserved across homologous sequences, including those from turtles and other vertebrate species.

For the g19.t1 gene, we validated the gene model predicted by AUGUSTUS and concluded that it was the most refined consensus model. We identified that the gene encodes a mitochondrial cholesterol side-chain cleavage enzyme (CYP11A1), as predicted by BLAST (Table 2) and InterPro (Fig 2A), and is important in C21-steroid hormone biosynthesis. Previous studies have shown that the cholesterol side-chain cleavage enzyme catalyzes the initial, rate-limiting step of steroidogenesis by cleaving cholesterol and generating pregnenolone—a precursor for essential steroids in the adrenal, gonads, and placenta such as progesterone, cortisol, and aldosterone [35, 36]. In addition to steroid production, the cholesterol side-chain cleavage enzyme also has been demonstrated to play a role in the morphology of the mitochondria, particularly its cristae shape which is important for oxidative phosphorylation [35]. This study agreed with our WoLF PSORT (Fig 3A) and TargetP (Fig 3E) results predicting a mitochondrial subcellular localization. Additionally, mutations in the *CYP11A1* gene can lead to life-threatening adrenal and gonadal insufficiencies, disruptions in the corticosterone circadian rhythm, and blunted stress response [36–38]. These findings highlight the evolutionary importance of the *CYP11A1* gene through its highly conserved sequence.

Through our analysis of the g112.t1 gene, we found that it likely encodes methylmalonyl-CoA epimerase (MCEE), an essential enzyme involved in branched-chain amino acid and odd-numbered fatty acid catabolism by converting (2R)-methylmalonyl-CoA to (2S)-methylmalonyl-CoA [39]. This pathway eventually leads to the downstream formation succinyl-CoA for the citric acid cycle. A previous paper studied the crystal structure of this enzyme and found that MCEE exists in a stable dimer with an 8-stranded beta sheet consisting of monomers folded into two tandem beta-alpha-beta-beta-beta modules [39], which was very consistent with the SWISS-MODEL homology model (Fig 8A and 8B). *MCEE* deficiency has been

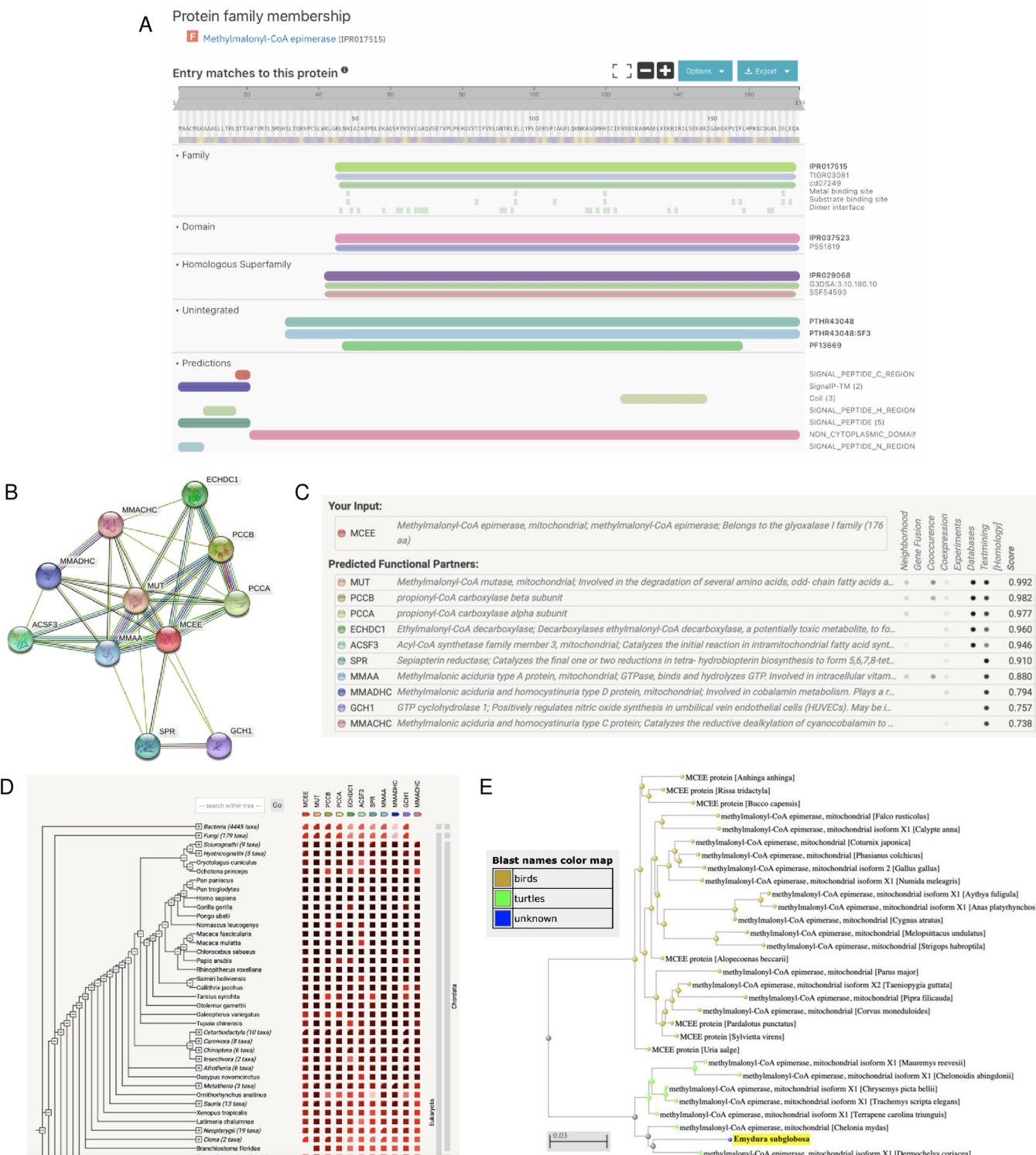

**Fig 6. Functional analysis of the MCEE enzyme.** (A) InterPro functional analysis of the enzyme. (B) STRING network of predicted protein-protein interactions in *H. sapiens*. (C) List of functional partners predicted by STRING corresponding to B. (D) Gene co-occurrence of the enzyme. (E) BLAST phylogenetic tree built based on pairwise alignment.

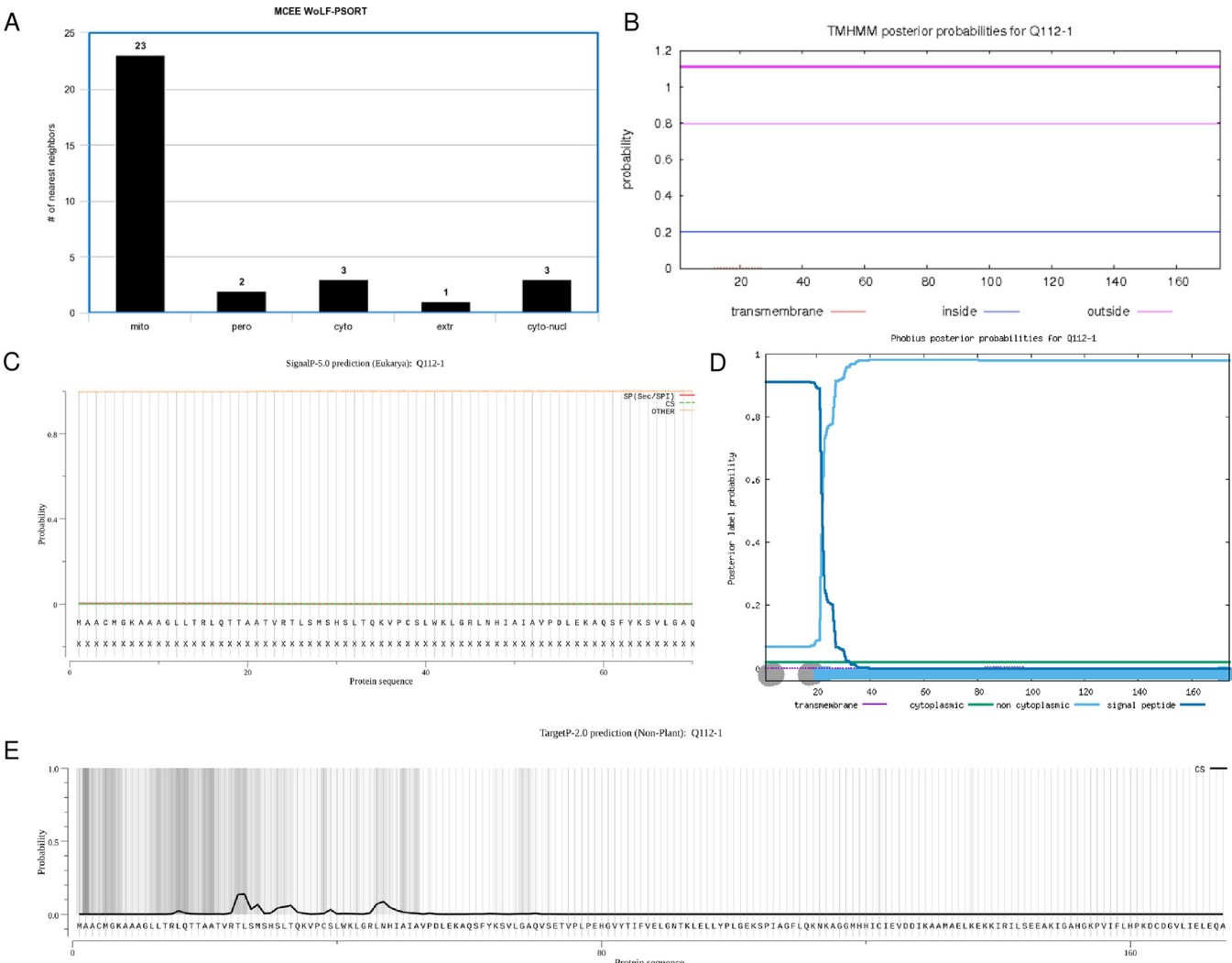

**Fig 7. Subcellular localization of the MCEE enzyme.** (A) Bar chart showing WoLF PSORT prediction of the protein's localization sites based on 32 nearest neighbors. Mito, mitochondria; pero = peroxisome; cyto = cytoplasm; extr = extracellular; cyto-nucl, cytoplasm and nucleus. (B) TMHMM prediction of TMHs. X-axis represents the amino acid number, and y-axis represents the probability that the amino acid is located within, outside, or inside the membrane. Probabilities >0.75 are significant. (C) SignalP analysis of signal sequences existing in the amino acid sequence of the polypeptide. (D) Phobius predictions of TMHs and signal peptides. X-axis represents the amino acid number, and y-axis represents the probability that the amino acid is transmembrane, cytoplasmic, non-cytoplasmic, and/or a signal peptide. Probabilities >0.75 are significant. (E) TargetP-2.0 prediction of N-terminal pre-sequences, signal peptides, and transit peptides.

shown to lead to methylmalonic aciduria [40] and central nervous system damage in humans [39]. Therefore, *MCEE* is an important gene for catabolism, so turtles can respond and adapt to their changing needs for energy.

With the study's focus on these two enzymes, it is important to note that they both are distantly related by the TCA cycle and play a role in the metabolic pathways of the mitochondria. With respect to the TCA cycle, MCEE acts upstream of succinyl-CoA [41], while CYP11A1 acts downstream of citrate [42]. Specifically, succinyl-CoA is a downstream product of the pathway converting propionyl-CoA to succinyl-CoA in which MCEE is involved [41]; citrate is an upstream input of the mevalonate metabolic pathway for cholesterol synthesis and eventual cleavage by CYP11A1 [42]. Future discussion of the spatial orientation of the two mitochondrial protein-coding genes (*CYP11A1* and *MCEE*) may be useful to study their

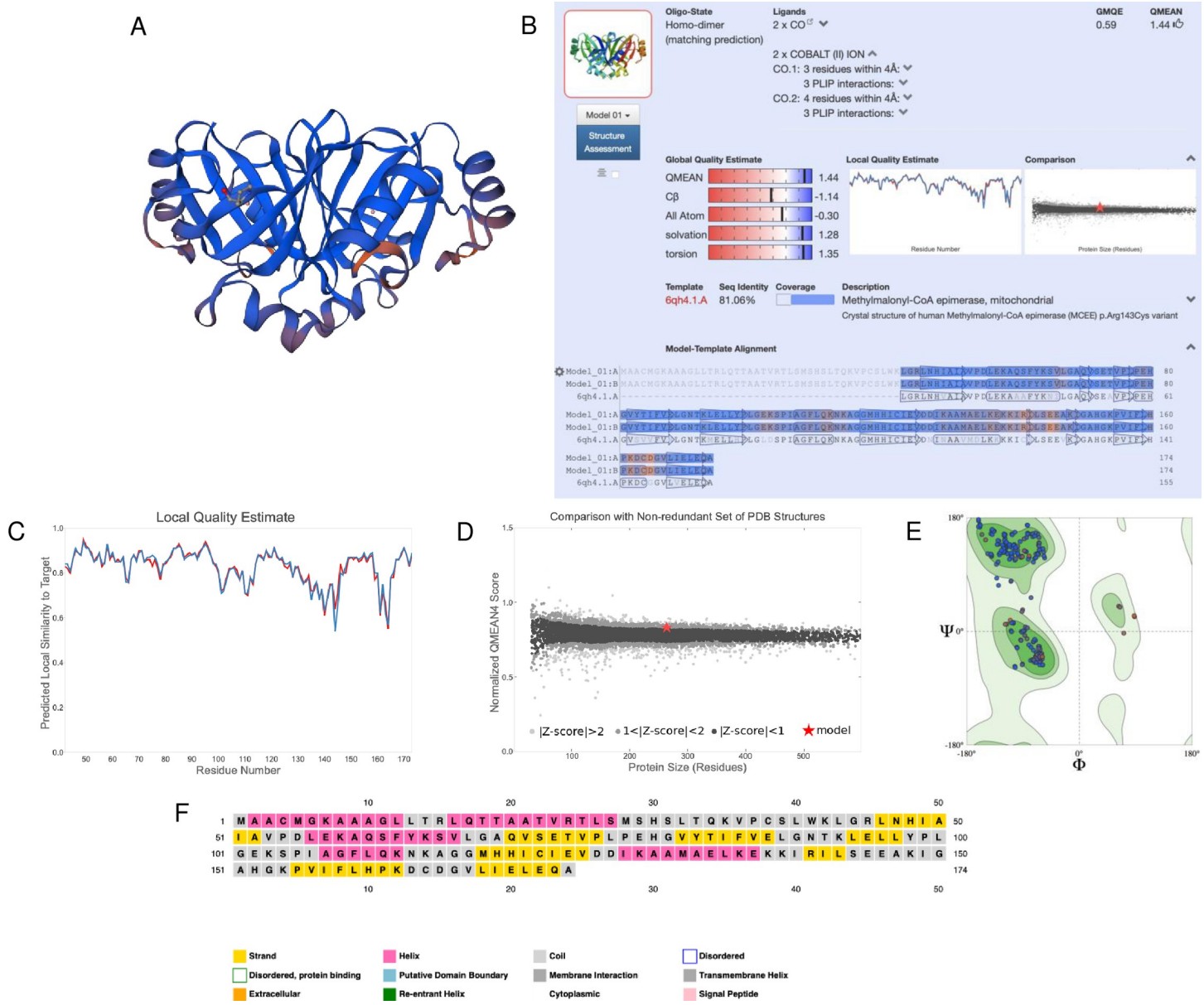

**Fig 8. Homology modeling and structural predictions of the MCEE enzyme.** (A) Three-dimensional homology model built by SWISS-MODEL. Blue regions are highly conserved, while orange regions are less conserved. (B) Oligo-state, ligands, global quality estimates, template, sequence identity, and coverage outputted by SWISS-MODEL. (C) Local quality estimate showing pair residue estimates. Similarities >0.6 are high-quality models. (D) Comparison with non-redundant set of PDB structures showing QMEAN scores for experimental structures that have been deposited of similar size. The red star is our model. (E) Ramachandran plot showing the probability of a residue having a specific orientation. Dots in the dark green regions represents high probability and a high-quality model. (F) Secondary structure prediction through PSIPRED.

transcriptional regulation and expression, especially since they are located on the same scaffold.

Elucidating the *E. subglobosa* genome through manual annotation contributes to the increasingly important field of conservation genomics and biodiversity [43]. With over half of the turtle species threatened by extinction, we characterized two highly conserved genes to further contextualize the reproductive biology of turtles and document their genetic diversity. The reliability of our study stems from the integration of multiple computational tools to

**Table 6. MolProbity results to validate the SWISS-MODEL prediction for MCEE.** A MolProbity Score close to 0 represents the resolution that one would see a structure of this quality. Clash score represents overlapping residues; a lower value is favored. Outliers represent values that extend outside the standard deviation; low values are also favored. Low values for bad bonds and angles are also favored.

| | |
|---|---|
| **MolProbity Score** | 1.4 |
| **Clash Score** | 5.07 |
| **Ramachandran Favoured** | 97.31% |
| **Ramachandran Outliers** | 0.77% |
| **C-Beta Deviations** | 1 |
| **Bad Bonds** | 0/2070 |
| **Bad Angles** | 13/2794 |

perform a holistic analysis on our genes of interest, which can serve as a tutorial for future studies involving structural and functional gene annotation. We utilized specific, complementary tools for various genomic analyses such as conservation, structure, function, localization, and phylogeny. RNA-seq experiments may help to further contextualize the *E. subglobosa* genome and contribute to our expanding knowledge about turtle biodiversity. The genes presented in this study contribute to the longevity phenotype of turtles and can be used by geneticists to direct conservation efforts towards critically endangered species. Furthermore, knowledge of the genes that are involved in the reproductive success and longevity of turtles can be further applied to clinical settings such as disease treatment in humans. Overall, our functional annotation of the *E. subglobosa* genome advances management and conservation efforts aimed to save currently endangered turtle and vertebrate species, provides clinical importance and application for human therapies, and adds significant progress towards a fully annotated genome.

## Supporting information

**S1 Fig. COBALT phylogenetic tree of the mitochondrial cholesterol side-chain cleavage enzyme.**
(TIF)

**S2 Fig. AmiGO 2 gene ontology graphs for the mitochondrial MCEE enzyme.** (A) Biological process L-methylmalonyl-CoA metabolic (GO:0046491). (B) Molecular function methylmalonyl-CoA epimerase (GO:0004493).
(TIF)

**S3 Fig. COBALT phylogenetic tree of the mitochondrial MCEE enzyme.**
(TIF)

**S1 Table. Sequences of putative proteins.**
(PDF)

## Acknowledgments

Thank you to Matteo Pellegrini, Leroy Bondhus, and Noah Alexander for their teaching and guidance. Thank you to Brad Shaffer for his edits.

## Author Contributions

**Conceptualization:** Megan Yu.

**Formal analysis:** Megan Yu.

**Investigation:** Megan Yu.

**Methodology:** Megan Yu.

**Project administration:** Megan Yu.

**Resources:** Megan Yu.

**Validation:** Megan Yu.

**Visualization:** Megan Yu.

**Writing – original draft:** Megan Yu.

**Writing – review & editing:** Megan Yu.

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
