## [Decision Letter · Decision Letter 0]

10 Mar 2022

PONE-D-21-26684Computational analysis on two putative mitochondrial protein-coding genes from the Emydura subglobosa genome: A functional annotation approachPLOS ONE

Dear Dr. Yu,

Thank you for submitting your manuscript to PLOS ONE. After careful consideration, we feel that it has merit but does not fully meet PLOS ONE’s publication criteria as it currently stands. Therefore, we invite you to submit a revised version of the manuscript that addresses the points raised during the review process.

We look forward to receiving your revised manuscript.

Kind regards,

Rakesh Kumar Verma, Ph.D

Academic Editor

PLOS ONE

2. Please note that in order to use the direct billing option the corresponding author must be affiliated with the chosen institute. Please either amend your manuscript to change the affiliation or corresponding author, or email us at plosone@plos.org with a request to remove this option.

Reviewers' comments:

Reviewer's Responses to Questions

**Comments to the Author**

1. Is the manuscript technically sound, and do the data support the conclusions?

Reviewer #1: Yes

2. Has the statistical analysis been performed appropriately and rigorously? 

Reviewer #1: Yes

3. Have the authors made all data underlying the findings in their manuscript fully available?

Reviewer #1: Yes

4. Is the manuscript presented in an intelligible fashion and written in standard English?

Reviewer #1: Yes

5. Review Comments to the Author

Reviewer #1: Thank you for asking me to review this paper and you are free to share my name with the authors if you wish.

I recommend this paper for publication, it is highly descriptive and highly detailed and well worthy of publication. A couple of comments below that are in my view minor issues and it should be good. The figures are appropriate and well done.

Cheers Scott

Scott A Thomson

Centro de Estudos dos Quelônios da Amazônia - CEQUA

Line 42: Compared to other eukaryotic species, turtles have minimal diversity (360 living species) and over 50% are threatened with extinction [2].

I find this sentence a little awkward and open to interpretation. “other eukaryotic species’ groups” of something similar may be better as it clarifies we you are talking about Orders in this case. Also historically there have been in the past many more species than are alive today there are many hundreds of fossil species, probably comparable with other reptilian orders.

In the materials and methods section the writing is as if there were multiple people involved here but there is a sole author. This may read better under the circumstances with some better use of third person writing rather than first person. This is just a suggestion the materials and methods is otherwise thorough.

Line 317: when you refer to “other turtle species” are you referring to previously studied taxa you mentioned in the intro? Such as C. n. abingdoni, G. gigantea and C. mydas? Maybe that should be clarified as E. subglobosa is to my knowledge the first Pleurodiran turtle to be studied this way, all others being Cryptodires, as such E. subglobosa being an outgroup or sister to all other studied turtles would be an expected result as at least 165 million years separates them. Addit. Ok I think I get it. It’s the species listed in your tables? you may want to make this clear somewhere early in the manuscript, ie material examined so the reader knows what your comparative material is.

6. PLOS authors have the option to publish the peer review history of their article (what does this mean?). If published, this will include your full peer review and any attached files.

Reviewer #1: **Yes: **Scott A. Thomson

---

## [Author Response · Author response to Decision Letter 0]

18 Apr 2022

Dear Dr. Rakesh Kumar Verma and Dr. Scott A. Thomson,

Thank you for your time and feedback in reviewing my original research article entitled “Computational analysis on two putative mitochondrial protein-coding genes from the Emydura subglobosa genome: A functional annotation approach.” I have addressed the reviewer’s comments and clarified specific phrases that may have been confusing to the reader:

Comment #1. Line 42: Compared to other eukaryotic species, turtles have minimal diversity (360 living species) and over 50% are threatened with extinction [2]. I find this sentence a little awkward and open to interpretation. “other eukaryotic species’ groups” of something similar may be better as it clarifies we you are talking about Orders in this case. Also historically there have been in the past many more species than are alive today there are many hundreds of fossil species, probably comparable with other reptilian orders.

I agree that talking about individual species may be ambiguous, as I am focusing on Orders in this sentence. I have changed the wording to “other eukaryotic species’ groups” to clarify this. 

Comment #2. In the materials and methods section the writing is as if there were multiple people involved here but there is a sole author. This may read better under the circumstances with some better use of third person writing rather than first person. This is just a suggestion the materials and methods is otherwise thorough.

This point is valid. I am used to writing in first-person, especially since most papers I have worked on have multiple authors. I have removed the first-person language and changed to third person instead. 

Comment #3. Line 317: when you refer to “other turtle species” are you referring to previously studied taxa you mentioned in the intro? Such as C. n. abingdoni, G. gigantea and C. mydas? Maybe that should be clarified as E. subglobosa is to my knowledge the first Pleurodiran turtle to be studied this way, all others being Cryptodires, as such E. subglobosa being an outgroup or sister to all other studied turtles would be an expected result as at least 165 million years separates them. Addit. Ok I think I get it. It’s the species listed in your tables? you may want to make this clear somewhere early in the manuscript, ie material examined so the reader knows what your comparative material is.

I have clarified this point specifically in the Materials and Methods section Lines 289-303. For “other turtle species,” I am referring to the species in the phylogenetic tree, which I obtained from the BLAST results. I expanded out the organism sets in the phylogenetic tree to contain species with lower percent identities in the BLAST results. I clarified “other turtle species” in Line 375-376 to “other turtle species in the phylogenetic tree.” Readers can refer to the Materials and Methods for how this tree was generated.

I have also reviewed the journal guidelines and ensured that my manuscript meets PLOS ONE’s style requirements. I confirm that I am affiliated with the University of California, Los Angeles (UCLA) Department of Molecular, Cell & Developmental Biology. I did not update any figures for this revision, and all figures were uploaded to PACE. All references are complete and correct. 

Please address correspondence to me via email at yumr247@g.ucla.edu. 

Thank you very much for your time and consideration of this manuscript. I look forward to hearing back from you and PLOS ONE. 

Sincerely,

Megan Yu

---

## [Editor Report · Decision Letter 1]

21 Apr 2022

Computational analysis on two putative mitochondrial protein-coding genes from the Emydura subglobosa genome: A functional annotation approach

PONE-D-21-26684R1

Dear Dr. Yu,

We’re pleased to inform you that your manuscript has been judged scientifically suitable for publication and will be formally accepted for publication once it meets all outstanding technical requirements.

Kind regards,

Rakesh Kumar Verma, Ph.D

Academic Editor

PLOS ONE
---

## [Editor Report · Acceptance letter]

10 Aug 2022

PONE-D-21-26684R1 

Computational analysis on two putative mitochondrial protein-coding genes from the *Emydura subglobosa* genome: A functional annotation approach 

Dear Dr. Yu:

I'm pleased to inform you that your manuscript has been deemed suitable for publication in PLOS ONE. Congratulations! Your manuscript is now with our production department. 

Kind regards, 

on behalf of

Dr. Rakesh Kumar Verma 

Academic Editor

PLOS ONE